# Estimation of the prevalence of substance use by wastewater-based epidemiology study in four cities of Guangdong, China

**Wei Wang[1,2], Dan Wang[1,2], Li Liu[3]*, Cunxi Qiu[1], Junyi Fan[1], Yuhan Jin[1]**

**1** Department of Drug Prohibition and Public Security, Criminal Investigation Police University of China, Shenyang, Liaoning, China, **2** Key Laboratory of Drug Control Technology in Liaoning Province, Shenyang, Liaoning, China, **3** Department of Chronic Disease Research Institute, Center for Disease Control and Prevention of Liaoning, Liaoning, China

* liuli_lncdc@163.com

## Abstract

### Introduction

The widespread use of illegal drugs and their associated problems have emerged as a significant public health concern. This study was conducted to estimate the consumption and prevalence of substance use in selected cities of Guangdong Province through wastewater-based epidemiology.

### Methods

We collected influent wastewater samples from 67 wastewater treatment plants across four cities of Guangdong from May 2023 to April 2024. The samples were analyzed using solid-phase extraction and liquid chromatography-mass spectrometry to identify 10 commonly used drugs and their metabolites in wastewater. By measuring the concentrations of these drug biomarkers, we estimated drug consumption, prevalence, and the number of individuals abusing drugs.

### Results

Our analysis revealed the presence of six out of ten monitored illicit drugs in the samples from the four cities. Methamphetamine emerged as the most consumed drug in Guangdong Province, with consumption ranging from 65 to 223 mg/1000 inh/d. This was followed by heroin (19–55 mg/1000 inh/d), codeine (7–20 mg/1000 inh/d) and ketamine (1–13 mg/1000 inh/d). The prevalence rates of methamphetamine, heroin, and ketamine across four cities of Guangdong Province were found to be 0.149%-0.411%, 0.003%-0.019%, and 0.003%-0.196%, respectively. Notably, between 2023 and 2024, the prevalence of heroin displayed a notable downward trend, while the prevalence of both methamphetamine and ketamine exhibited a marked upward trend.

**Data availability statement:** All relevant data are within the paper and its Supporting Information files.

**Funding:** This study was supported by the National Key R&D Program Project of China (No. 2018YFC0807405), the Guangdong Provincial Public Security Department Drug Abuse Scale Investigation Project (No. 2019-54). The funders had no role in study design, data collection and analysis, decision to publish, or preparation of the manuscript.

## Discussion

Our comprehensive analysis of the substance use situation in these cities indicated that methamphetamine, heroin, and ketamine were the most used substances. Cocaine was detected in only two WWTPs whereas MDMA was found in two separate plants. The cities with the highest and lowest prevalence rates of these three drugs were variant. This study provides valuable data that can support real-time monitoring of regional substance use situations, aiding in developing effective intervention strategies.

## 1. Introduction

The use of illegal drugs and associated issues have emerged as significant public health concerns in numerous regions globally [1–3]. Substance use not only elevates the risk of developing additional mental health disorders but also heightens the likelihood of transmitting infectious diseases such as hepatitis C and HIV/AIDS [4]. Furthermore, substance use is associated with antisocial and aggressive behavior and is linked to criminal activities, which substantially threaten public safety on a global scale[5]. Consequently, numerous developed countries have made substantial investments in the development of monitoring systems to assess the extent and implications of substance use [6, 7]. Population-based surveys and reporting systems faced limitations due to issues such as inadequate sample representativeness and underreporting of cases, rendering them unable to accurately gauge the extent of substance use across the entire population [8].

The wastewater-based epidemiology (WBE) concept was first introduced by American environmental scientist Daughton [9] in 2001. Later, in 2005, Zuccato [10] pioneered its application to track cocaine usage in Italy, and found that by using domestic sewage to detect the concentration of cocaine and its metabolites, the estimated per capita consumption of cocaine was highly consistent with the data from previous surveys. Over recent years, this field has garnered significant scholarly attention, leading to its widespread utilization in detecting environmental exposure substances within wastewater. This is primarily due to wastewater's intricate composition, encompassing various chemical and biological markers stemming from human activities. By analyzing the concentration of specific substances in wastewater, insights can be gained into the consumption patterns of legal drugs [11], illegal drugs [12], tobacco [13], and alcohol [14] by residents within the service area of wastewater treatment plants (WWTPs). In the last decade, both wastewater analysis and WBE have been established as supplementary methods for assessing the prevalence of illegal substance use among populations [15–17]. This method has gained widespread international recognition, offering to provide objective data for an investigation into illicit substance use in the region. The use of illegal drugs, such as heroin, cocaine, methamphetamine, and ketamine has been a long-standing concern for scientists across various countries [18,19]. Notably, the 2021 World Drug Report acknowledged the findings from WBE's drug monitoring studies conducted in several pertinent countries. Most of these findings align with the data presented in the World Drug Report [20], underscoring the efficacy of this approach in providing precise, uninterrupted, and reliable information on drug usage. Furthermore, it offers a cost-effective, convenient, and real-time means of monitoring drug trends, thereby addressing the limitations inherent in traditional methods, such as time delays and data constraints.

This study focused on Guangdong Province, a region in China that faces a comparatively severe substance use situation. We gathered wastewater samples from 67 WWTPs in four cities within the province. By using a WBE approach, we assessed the magnitude of the

substance use population in these four selected cities. Our objective was to conduct a scientific evaluation of the scale of the consumption of substance use in Guangdong, in order to strategically plan the allocation of health resources and devise effective addiction prevention plans, and to establish a substance use monitoring program based on wastewater-based epidemiology to detect the temporal trends of major used substances, and to prevent substance use.

## 2. Materials and methods

### 2.1. Reagents and materials

Standard solutions of ten target analytes namely morphine, 6-acetylmorphine, methamphetamine, amphetamines, ketamine, norketamine, MDMA, MDA, cocaine, and benzoyleconazole along with their respective internal standards (including Morphine-D3, 6-acetylmorphine-D3, methamphetamine-D5, amphetamine-D5, ketamine-D4, norketamine-D4, MDMA-D5, MDA-D5, cocaine-D3, benzoylaconitine-D3 deuterated analogs) were purchased from Cerilliant (Round Rock, TX, USA). Laboratory ultrapure water was generated using Milli-Q system (Millipore, USA), hydrochloric acid and ammonia were sourced from Beijing Bailingwei Technology Co., Ltd. (Beijing, China). Formic acid was obtained from Sigma Aldrich (Germany). Additionally, methanol and acetonitrile (both chromatographic grade) were purchased from Fisher Scientific (USA). Oasis MCX SPE cartridges (60mg, 3mL) were acquired from Waters Corporation (Milford, MA, USA). Other materials include glass fiber filter membrane (GF/F 0.7 mm, Whatman UK), needle-type nylon microporous filter membrane (13 mm x 0.22μm, Tianjin Jinteng Company), and disposable sterile needle syringe (Jiangsu Zhiyu Medical Equipment Co., Ltd.).

### 2.2. Sample collection

In May 2023 and April 2024 respectively, a stratified random sampling method was employed, stratifying by region and economic level, to select one city from each of the Pearl River Delta region, northern, eastern, and western parts of Guangdong Province: Guangzhou, Qingyuan, Shantou, and Maoming, respectively. Wastewater samples were gathered from 67 WWTP inlets across these four cities. An automatic water quality sampler was utilized to collect the samples from each plant. The sampling procedure involved collecting 50 mL of wastewater every 0.5 hours in a time-proportional mode, continuously for 24 hours, resulting in a total of 500 mL of untreated wastewater being collected at each plant. To prevent the biodegradation of toxic substances in sewage, 2ml of hydrochloric acid was added to each sample. The samples were then transported to the laboratory via cold chain to ensure their integrity. Upon arrival, the samples were stored at -40°C and underwent pre-treatment within 24 hours. This study has received approval from the Ethics Committee of the School of Narcotics Control and Public Security, Criminal Investigation Police University of China (2022-12).

### 2.3. Sample analysis

Sample pretreatment involved solid-phase extraction (SPE) using mixed reversed-phase/cation-exchange cartridges (Oasis-MCX), as detailed by Castiglioni [21]. First, 50mL of filtrate from the sewage sample was transferred to a centrifuge tube with a stopper. Then, 100μl of mixed deuterium internal standard at 25 ng/mL was added. The mixture was thoroughly agitated and subsequently transferred to an activated SPE column. The column was rinsed with 4mL of methanol at a controlled flow rate of 4.0mL/min. Next, the SPE column was centrifuged or vacuum-dried until completely dry. It was then washed with 4mL of 5% ammonia methanol solution at a flow rate of 1.0mL/min, and the eluent was collected. The eluent was

concentrated at 60 °C on a concentrator until almost dry, after which 250 μL of 0.1% formic acid aqueous solution was added. The mixture was well-blended, filtered through a water system microporous filter membrane, and used as the extraction solution for the test sample for instrumental analysis. For each test sample, two parallel extraction solutions were independently prepared.

The target compounds were separated using liquid chromatography-mass spectrometry (LC-MS) on a Waters Xevo TQ-S system equipped with ACQUITY UPLC BEH C18 column. The injection volume was 5μL. The mobile phases consisted of 0.1% formic acid aqueous solution (phase A) and 0.1% formic acid acetonitrile solution (phase B). The elution gradient was as follows: 0–6.0min: 95% A; 6.0–6.2 minutes: 75% A; 6.2–8.0 minutes: 0% A; 8.0–8.2min: 0% A; 8.2–11.0min: 95%.

Electrospray ionization was operated in positive ionization mode. The mass spectrometry (MS) system was set to multi-reaction monitoring (MRM) mode for quantification. The electrospray spray voltage (IS) was 3000 V; the curtain air pressure (CUR) was 30 Psi (1 Psi = 6.89 kPa); the collision air pressure (CAD) was 8 Psi; the nebulizer gas pressure (GSI) was 55 Psi; the auxiliary heating pressure (GS2) was 65 Psi; the auxiliary gas temperature was 550 °C. The delustering potential, collision energy, quantifier, qualifier ions, and retention time of each compound are shown in Table 1.

## 2.4. Quality assurance and quality control (QAQC)

To ensure the precision of our experiment and the trustworthiness of our study, we utilized multiple 50ml portions of drinking water. We introduced varying concentrations of mixed standard working solution, along with 100 μL of a 25 ng/mL mixed internal standard working solution. This allowed us to achieve target drug concentrations of 1 ng/L, 5 ng/L, 10 ng/L, 50 ng/L, 100 ng/L, 150 ng/L, 200 ng/L, and 250 ng/L. Our findings revealed that ten target

**Table 1. Qualitative and quantitative ion pairs and collision energy conditions for 10 detection target drugs.**

| Analyte | Retention time (min) | Declustering Potential (V) | Quantitative ion pair (m/z) | Qualitative ion pair (m/z) | Collision energy (eV) |
|---|---|---|---|---|---|
| Morphine | 1.2 | 120 | 286.1/201.1 | 286.1/201.1 | 36 |
| | | | | 286.1/165.1 | 57 |
| 6-Aetylmorphine | 3.1 | 90 | 328.2/165.1 | 328.2/211.1 | 36 |
| | | | | 328.2/165.1 | 52 |
| Amphetamine | 2.9 | 40 | 136.1/91.1 | 136.1/119.1 | 13 |
| | | | | 136.1/91.1 | 23 |
| Methamphetamine | 3.3 | 30 | 150.1/91.1 | 150.1/119.1 | 16 |
| | | | | 150.1/91.1 | 27 |
| Ketamine | 4.6 | 40 | 238.1/125.0 | 238.1/207.1 | 21 |
| | | | | 238.1/125.0 | 39 |
| Norketamine | 4.3 | 40 | 224.1/125.0 | 224.1/207.1 | 18 |
| | | | | 224.1/125.0 | 35 |
| MDMA | 3.5 | 35 | 194.1/163.1 | 194.1/163.1 | 18 |
| | | | | 194.1/105.1 | 34 |
| MDA | 3.1 | 40 | 180.1/133.1 | 180.1/133.1 | 25 |
| | | | | 180.1/105.1 | 30 |
| Benzoylecgonine | 4.2 | 70 | 290.1/168.1 | 290.1/168.1 | 28 |
| | | | | 290.1/105.0 | 41 |
| Cocaine | 6.7 | 60 | 304.2/182.1 | 304.2/182.1 | 28 |
| | | | | 304.2/150.1 | 34 |

drugs exhibited strong linearity. Specifically, morphine demonstrated a robust linear correlation between 5–250 ng/L, while the other drug targets showed a similar correlation between 1–250 ng/L, with $R^2 > 0.998$.

For our analytical samples, we prepared drinking water with concentrations of 0.25, 0.5, 1, 2.5, and 5 ng/L. Each concentration underwent ten parallel measurements. Referring to the qualitative evaluation standards [22], we determined the limits of detection (LOD) based on the mass concentration with a 100% positive detection rate. The limits of quantification (LOQ) were established using the minimum concentration point with an RSD below 15%, which was included in fitting the standard curve.

To ascertain the recovery rate, we employed the matrix spiked method. This involved adding 25 ng of the target drug to a 50 mL low-concentration water sample. The LOD, LOQ, and recovery rate for each target drug were detailed in Table 2. The actual spiked recovery rate in wastewater ranged between 82.8% and 98.7%, satisfying the methodological requirements. Furthermore, we performed repeatability experiments at five concentration points. The relative standard deviation (RSD) for all compound concentrations fell within the range of 1.49% to 7.39%, indicating excellent stability of the method and instrument.

## 2.5. Consumption and prevalence calculation

The target drug's load, expressed as milligrams per 1000 inhabitants per day (mg/1000 inh/d), refers to the daily mass load of each drug residue per 1000 inhabitants at a specific WWTP. This calculation is outlined in Equation (1). Within this formula, $C_i$ represents the concentration of drugs or their metabolites in the inflowing wastewater (ng/L), and $F_{In}$ denotes the daily inflow rate of the WWTP (L/day). Lastly, PS signifies the population served the plant, specifically expressed in increments of 1000 inhabitants.

$$\text{Load} = \frac{C_i \times F_{In}}{\dfrac{PS}{1000}} \times \frac{1}{10^6} \tag{1}$$

The target drug's consumption (mg/1000 inh/d), reflects the average drug intake of each drug residue among every 1000 people served by a specific WWTP. This calculation is detailed in Equation (2). Within this formula, $EF_i$ signifies the excretion rate of the target drug post-human metabolism (%), $MW_{pi}$ represents the relative molecular weight of the target drug (g/mol), and $MW_{mi}$ denotes the relative molecular weight of the target drug biomarker

**Table 2. The limit of detection (LOD), limit of quantification (LOQ), and recovery rate of the target drug.**

| Analyte | LOD (ng/L) | LOQ (ng/L) | Recovery rate (%) |
|---|---|---|---|
| Morphine | 0.5 | 1.0 | 84.8 ± 4.0 |
| 6-Aetylmorphine | 0.3 | 1.0 | 89.3 ± 1.9 |
| Amphetamine | 0.5 | 1.0 | 98.7 ± 5.5 |
| Methamphetamine | 0.5 | 1.0 | 96.7 ± 1.0 |
| Ketamine | 0.5 | 1.0 | 95.1 ± 1.1 |
| Norketamine | 0.3 | 1.0 | 93.6 ± 1.3 |
| MDMA | 0.5 | 1.0 | 95.1 ± 0.3 |
| MDA | 0.3 | 1.0 | 92.1 ± 2.2 |
| Benzoylecgonine | 0.3 | 1.0 | 82.8 ± 6.0 |
| Cocaine | 0.3 | 1.0 | 88.9 ± 3.7 |

(g/mol). By utilizing the population served by the WWTP as a weighting factor, we compute the average drug consumption per capita in a given city through a weighted average approach.

$$\text{Consumption} = \text{load} \times \frac{MW_{pi}}{MW_{mi}} \times \frac{1}{EF_i} \qquad (2)$$

The prevalence of a targeted drug refers to the percentage of individuals consuming a specific drug within a given region's total population. This proportion can be determined by calculating the consumption of the target drug, adjusting for specific age groups, and dividing by the daily drug dosage, as outlined in Equation (3).

$$\text{Prevalence} = \frac{m}{R \times D \times n \times 1000} \times 100\% \qquad (3)$$

Within this formula, m represents the consumption of the target drug in a specific region (mg/1000inh/d). R denotes the percentage of the population aged between 15 and 64 years old relative to the total population (%). Additionally, n signifies the frequency of drug consumption expressed in times per day, while D indicates the daily drug intake or consumption dose per individual (mg/inh/day). Referring to the 2020 National Population Census data of the People's Republic of China, the proportions of the population aged 15–64 in Guangzhou, Qingyuan, Shantou, and Maoming are 77.54%, 65.05%, 67.05%, and 60.84%, respectively. Furthermore, Table 3 presents the human excretion factors specific to the target drug, the ratio of Wpi/MWmi for the target drug, the frequency of consumption, and the daily drug intake or consumption dose per person.

## 2.6. Statistical analysis

All calculations and statistical analyses were performed using SPSS 22.0 (IBM Corporation, Armonk, NY, USA), with a significance level set at $P < 0.05$. Normality was assessed using the Shapiro–Wilk, skewness, and kurtosis tests. Correlation analysis between the amount of used substances and various variables was conducted using Pearson's and Spearman's correlation coefficients. A one-way ANOVA was employed to explore significant differences in overuse among different cities.

## 3. Results and discussion

### 3.1. Drug residue concentrations

In this study, eight out of 10 target drugs were identified in 67 WWTP samples across four cities in Guangdong, China (Table 4, Fig 1, and Table S2). The detection rates varied, with

**Table 3. The biomarker, excretion rate, correction factor, consumption frequency and daily drug intake/consumption dose from references.**

| Drug | Selected Biomarker | Excretion Factor (%) | MWp/MWm | The consumption frequency | The daily drug intake/consumption dose per person (mg/inh/day) | Reference |
|---|---|---|---|---|---|---|
| Heroin | 6-monoacetylmorphine | 42 | 1.3 | 2.4 | 15 | [23,24] |
| Codeine | Morphine | 6 | 1.1 | 2.4 | 38 | [23,25] |
| Methamphetamine | Methamphetamine | 43 | 1.0 | 0.3 | 30 | [16,23,26] |
| Ketamine | Ketamine | 16 | 1.0 | 0.4 | 75 | [24,27] |
| MDMA | MDMA | 26 | 1.0 | 0.4 | 100 | [23,28,29] |
| Cocaine | Benzoylecgonine | 29 | 1.1 | 0.3 | 100 | [21,23] |

**Table 4. Drug residue concentrations (ng/L) of WWTPs in four cities of Guangdong, 2023 to 2024.**

| Drug Residues | Guangzhou | | Qingyuan | | Shantou | | Maoming | |
|---|---|---|---|---|---|---|---|---|
| | 2023 | 2024 | 2023 | 2024 | 2023 | 2024 | 2023 | 2024 |
| Morphine | 12 ± 10 | 10 ± 6 | 18 ± 13 | 16 ± 12 | 16 ± 12 | 13 ± 11 | 7 ± 5 | 6 ± 5 |
| 6-AMA | 2 ± 1 | 1 ± 1 | 3 ± 1 | 2 ± 2 | 3 ± 1 | 3 ± 2 | 1 ± 1 | 1 ± 1 |
| Amphetamine | 5 ± 4 | 7 ± 5 | 5 ± 4 | 6 ± 5 | 5 ± 3 | 6 ± 3 | 3 ± 2 | 4 ± 3 |
| Methamphetamine | 105 ± 77 | 117 ± 82 | 85 ± 23 | 96 ± 28 | 101 ± 21 | 108 ± 26 | 66 ± 43 | 66 ± 43 |
| Ketamine | 11 ± 8[a] | 12 ± 8[a] | 18 ± 13[a] | 20 ± 16[a] | 92 ± 51[a] | 99 ± 63[a] | 3 ± 2[a] | 6 ± 4[a] |
| Norketamine | 0.4 ± 0.1[b] | 0.6 ± 0.5[b] | 0.3 ± 0.1[b] | 0.5 ± 0.4[b] | 5 ± 3 | 7 ± 5 | 0.1 ± 0.1[a b] | 0.4 ± 0.2[b] |
| MDMA | – | – | 0.4 ± 0.1 | – | – | – | – | – |
| MDA | – | – | – | – | – | – | – | – |
| Benzoylecgonine | – | – | – | – | – | – | 0.3 ± 0.1 | 0.5 ± 0.4 |
| Cocaine | – | – | – | – | – | – | – | – |

Note: All values are mean ± SD of three replicates: a Means followed by different superscript alphabets in each line are significantly different (P < 0.05) among different cities. b Means followed by different superscript alphabets in each line are significantly different (P < 0.05) among different years.

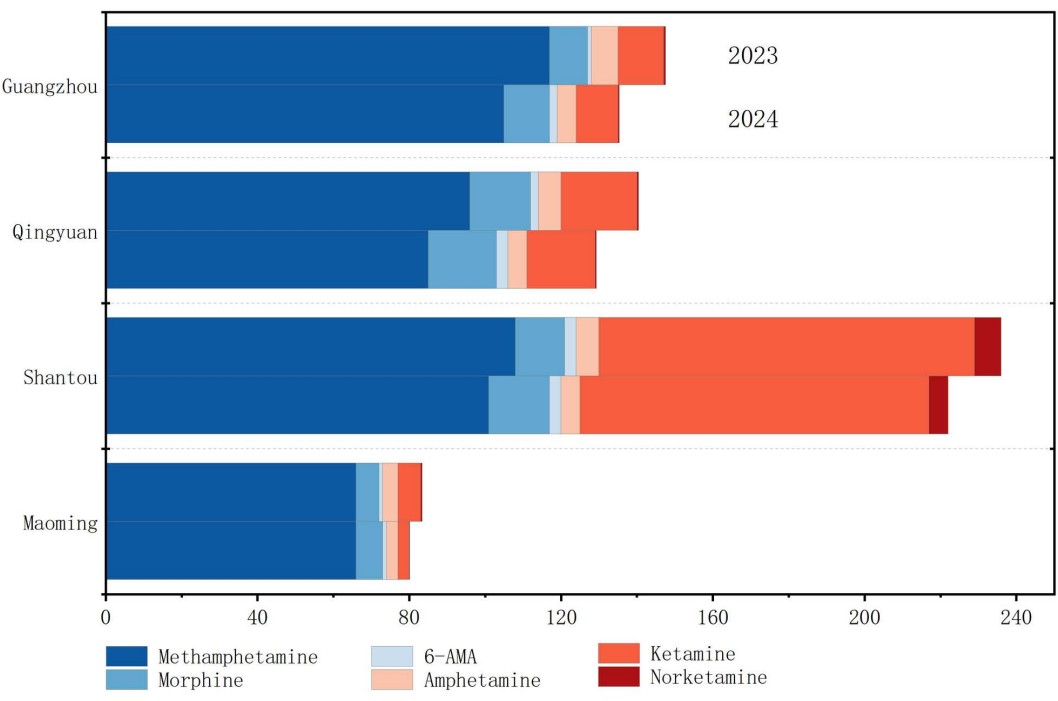

**Fig 1. Drug residue concentrations (ng/L) of WWTPs in four cities of Guangdong, 2023 to 2024.**

6-MAM, a specific metabolite of heroin that serves as a reliable indicator of heroin use, detected in 54% of the samples, and morphine, which has multiple sources including heroin metabolits and other drugs like analgesic morphine, codeine, ethylmorphine, and fluco-nazole, was detected in 93% of the samples. The detection rates ranged from 40% to 94% for amphetamine, methamphetamine, norketamine, and ketamine. Benzoylecgonin (a metabo-lite of cocaine) was only found in two WWTPs in Maoming, while MDMA was detected in two plants in Shantou. Notably, a strong positive correlation (r = 0.84, p < 0.01) was observed

between the concentrations of amphetamine and methamphetamine in wastewater samples, possibly due to the degradation of methamphetamine into amphetamine in both humans and wastewater. The ratio of amphetamine and methamphetamine concentrations in the samples was mostly at or below 20%, suggesting that the amphetamine present primarily originated from methamphetamine metabolism [30]. When comparing across cities, it was observed that the concentrations of methamphetamine in Guangzhou's wastewater were notably higher than in the other three cities, indicating a potential positive correlation between economic development level and drug-related issues [30]. Notably, the ratio of ketamine concentration in each WWTP to the concentration of ketamine after human metabolism ranges between 6.8 and 100. This is significantly higher than the ratio of residual ketamine to its main metabolite (0.27–2.1) [31]. This discrepancy may be attributed to the direct disposal of ketamine, leading to an overall elevated level of ketamine in the wastewater compared to that resulting from human metabolism.

## 3.2. Estimation of drug consumption

As shown in Table 5 and Fig 2, methamphetamine exhibited the highest estimated per capita consumption in four cities of Guangdong, ranging from 65 to 223 mg/1000 inh/d. This was followed by heroin, with consumption ranging from 19 to 55 mg/1000 inh/d, codeine, with consumption ranging from 7 to 20 mg/1000 inh/d, and ketamine, with the lowest consumption of 1–13 mg/1000 inh/d. Through a one-way ANOVA analysis, we observed significant statistical differences in the per capita consumption of various drugs across these cities. Specifically, for the year 2023, the results showed significant differences for heroin ($F = 5.85$, $P < 0.05$), methamphetamine ($F = 3.58$, $P < 0.05$), codeine ($F = 2.96$, $P < 0.05$), and ketamine ($F = 6.85$, $P < 0.05$). Similarly, in 2024, significant statistical differences were again observed for heroin ($F = 6.83$, $P < 0.05$), methamphetamine ($F = 2.95$, $P < 0.05$), codeine ($F = 2.91$, $P < 0.05$), and ketamine ($F = 6.54$, $P < 0.05$) across these cities.

In this study, the estimated consumption of methamphetamine stood out as the highest estimated consumption. When compared to other studies, the consumption of methamphetamine in Guangdong Province surpassed that of Beijing, Shanghai, and Shenzhen, but remained below the levels in Turkey and various European cities [31–34]. The per capita consumption levels of heroin in Guangdong were notably lower than the average consumption level reported in China (260 ± 116 mg/1000 inh/day) [32], Turkey (1000 mg/1000 inh/day) [31] and the United States (474 ± 32 mg/1000 inh/day) [35]. The per capita consumption levels of codeine in Guangdong were similar to the average consumption level reported in China (1–37 mg/1000 inh/day) [36], New Zealand (57 mg/1000 inh/day) [37] and the United States (140 mg/1000 inh/day) [38]. As for ketamine, the per capita consumption

**Table 5. Consumption (mg/1000 inh/d) of substance use in four cities of Guangdong province, 2023 to 2024.**

| City | Heroin consumption | | Codeine consumption | | Methamphetamine consumption | | Ketamine consumption | |
|---|---|---|---|---|---|---|---|---|
| | 2023 | 2024 | 2023 | 2024 | 2023 | 2024 | 2023 | 2024 |
| Guangzhou | 28 ± 21[a] | 24 ± 18[a] | 12 ± 10[a] | 11 ± 9[a] | 216 ± 192[a] | 223 ± 199[a] | 9 ± 7[a] | 10 ± 7[a] |
| Shantou | 20 ± 14[a] | 19 ± 12[a] | 9 ± 6[a] | 7 ± 5[a] | 88 ± 64[a] | 100 ± 88[a] | 1 ± 1[a] | 2 ± 1[a] |
| Qingyuan | 43 ± 21[a] | 38 ± 17[a] | 16 ± 12[a] | 14 ± 10[a] | 174 ± 130[a] | 196 ± 145[a] | 10 ± 8[a] | 13 ± 11[a] |
| Maoming | 55 ± 38[a] | 52 ± 35[a] | 20 ± 13[a] | 18 ± 12[a] | 65 ± 38[a] | 82 ± 44[a] | 3 ± 3[a] | 5 ± 4[a] |

Note: All values are mean ± SD of consumption: a Means followed by different superscript alphabets in each row are significantly different ($P < 0.05$) among different cities.

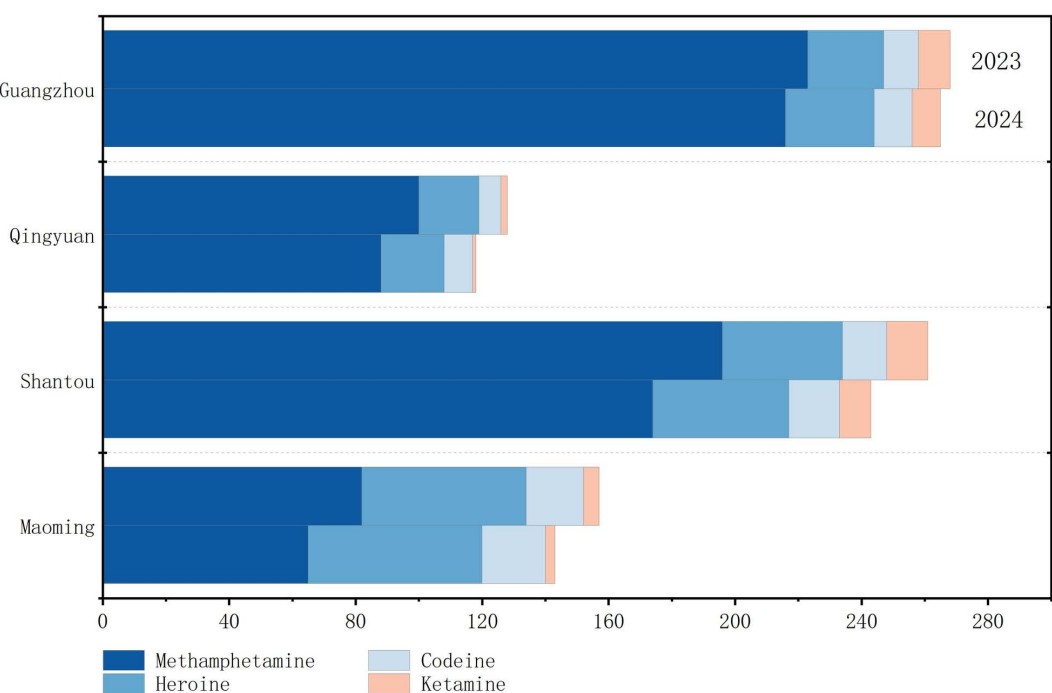

**Fig 2. Consumption (mg/1000 inh/d) of drugs of abuse in four cities of Guangdong province, 2023 to 2024.**

levels in Guangdong were similar to the national average level in China (12 mg/1000 inh/day) [32], yet significantly lower than those in Malaysia (357 mg/1000 inh/day) [39] and Canada (195 mg/1000 inh/day) [16].

### 3.3. Estimation of the prevalence of three major substance use

Our monitoring study estimated the prevalence of three primary used substances, specifically heroin, methamphetamine, and ketamine. The findings, as presented in Table 6, indicate that the prevalence rates for methamphetamine, heroin, and ketamine across four cities in Guangdong Province were 0.149%-0.347%, 0.004%-0.019%, and 0.003%-0.152%, respectively, in 2023. In 2024, these rates shifted to 0.152%-0.411% for methamphetamine, 0.003%-0.016% for heroin, and 0.013%-0.196% for ketamine. Notably, the prevalence of heroin exhibited a downward trend, whereas the prevalence of both methamphetamine and ketamine demonstrated an upward trend. The World Drug Report 2023 [40], released by the United Nations Office on Drugs and Crime, highlights that methamphetamine was likely the most extensively used and supplied synthetic drug globally, approximately 36 million

**Table 6. Estimation of the prevalence (%) of three major substance use in four cities of Guangdong province.**

| City | Heroin | | Methamphetamine | | Ketamine | |
|---|---|---|---|---|---|---|
| | 2023 | 2024 | 2023 | 2024 | 2023 | 2024 |
| Guangzhou | 0.017 ± 0.004 | 0.015 ± 0.003 | 0.347 ± 0.224 | 0.378 ± 0.232 | 0.049 ± 0.032 | 0.067 ± 0.039 |
| Shantoun | 0.004 ± 0.001 | 0.003 ± 0.001 | 0.149 ± 0.103 | 0.168 ± 0.127 | 0.005 ± 0.003 | 0.019 ± 0.008 |
| Qingyua | 0.019 ± 0.006 | 0.016 ± 0.004 | 0.370 ± 0.247 | 0.411 ± 0.265 | 0.152 ± 0.084 | 0.196 ± 0.092 |
| Maoming | 0.010 ± 0.003 | 0.007 ± 0.002 | 0.126 ± 0.082 | 0.152 ± 0.086 | 0.003 ± 0.001 | 0.013 ± 0.006 |

individuals had used amphetamines in the preceding year. When compared to other WBE studies, the prevalence of methamphetamine in the four cities of Guangdong province was similar to that in Qinghai-Tibet Plateau (ranging from 0.003–0.197) [36], but notably lower than in 22 Chinese cities(0.18–1.25%)[35]. However, the prevalence of ketamine is lower than that of Beijing (0.0210%, 0.0710%) [41] and the Qinghai-Tibet Plateau(0.011–0.044) [36]. In summary, the prevalence of synthetic drugs in Guangdong Province occupied a relatively intermediate level.

In this study, the limitations of WBE are manifested in several ways. Firstly, the absence of typical dosage, drug type, purity, and other indicators of drug users in wastewater epidemiology, coupled with the fact that wastewater samples can only be tested instantaneously, poses a challenge in representing annual data variations. To accurately predict the magnitude of substance use, prolonged monitoring and evaluation are essential. Secondly, the dearth of research on metabolites' stability and metabolic rate in synthetic drugs obstructs accurate evaluation. Finally, the target substance may be adsorbed by the pipeline and undergo reactions in the natural environment, leading to inaccurate results. Therefore, it is imperative to integrate WBE with comprehensive and detailed social epidemiological investigations to obtain more precise evaluation outcomes.

## 4. Conclusions

Utilizing a wastewater-based epidemiology (WBE) approach, this study scientifically evaluates the substance use population's consumption scale and incidence trend in four selected cities of Guangdong Province, China. Among the ten target drugs, six were identified in WWTPs from these cities. The primary used substance in the four cities of Guangdong Province is methamphetamine, with heroin and ketamine following closely behind. Our survey revealed that the prevalence of synthetic drugs in Guangdong Province stands at a relatively intermediate level, indicating significant progress in the fight against substance use in Guangdong, China. This study serves as a valuable guide for future WBE monitoring efforts in China, enabling more effective planning of health resource allocation and establishing a substance use monitoring program based on wastewater-based epidemiology to detect the temporal trends of major used substances, and to prevent substance use.

The following are available online. S1 Table Information on WWTPs in four Cities in Guangdong Province. S2 Table Concentration values of target drugs detected in all WWTPs. S3 Table Mean influent loads (mg/1000 inh/d) of drug residues in four cities of Guangdong.

## Supporting information

**S1 Table. Information on WWTPs in Four Cities in Guangdong Province.**
(DOCX)

**S2 Table. Concentration values of target drugs detected in all WWTPs.**
(DOCX)

**S3 Table. Mean influent loads (mg/1000 inh/d) of drug residues in four cities of Guangdong, 2023–2024.**
(DOCX)

## Acknowledgments

We are grateful to many local investigators from Guangzhou, Shantou, Qingyuan, and Maoming in the Guangdong Province of China for their assistance with data collection and to other staff who participated in this research project.

## Author contributions

**Data curation:** Li Liu, Junyi Fan, Yuhan Jin.

**Investigation:** Li Liu.

**Methodology:** Wei Wang, Dan Wang, Cunxi Qiu.

**Supervision:** Dan Wang.

**Writing – original draft:** Cunxi Qiu, Junyi Fan, Yuhan Jin.

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
