## [Decision Letter · Decision Letter 0]

28 Jan 2025

PONE-D-24-60707Estimation of the prevalence of drug abuse by wastewater-based epidemiology study in four cities of Guangdong, ChinaPLOS ONE

Dear Dr. Wang,

Thank you for submitting your manuscript to PLOS ONE. After careful consideration, we feel that it has merit but does not fully meet PLOS ONE’s publication criteria as it currently stands. Therefore, we invite you to submit a revised version of the manuscript that addresses the points raised during the review process.

We look forward to receiving your revised manuscript.

Kind regards,

David Wampler

Academic Editor

PLOS ONE

“This research was funded by the National Key R&D Program Project (No. 2018YFC0807405)， the Guangdong Provincial Public Security Department Drug Abuse Scale Investigation Project (No. 2019-54).”

Additional Editor Comments:

Dear Authors,

Congratulations, the reviews for the first round of peer review were very positive, and I am sure that with some minor revision your manuscript will be a great fit for PLOS One.

I would like to thank you for your contribution.

I have included the reviewer comments below, in addition to a few editorial comments.

Respectfully,

David Wampler, PhD, LP, FAEMS

Academic Editor

Editor notes:

Every figure and table should be able to stand on its own, please make sure that all acronyms and abbreviations are defined in the table or figure legend.

I would recommend not using acronyms in the first sentence of the conclusions - this sentence is most commonly quoted, reduce confusion if you use full names.

The first sentence of the conclusion is a bit weak, your real findings are presented on second sentence. consider strengthening the first sentence to include your actual findings.

Reviewers' comments:

Reviewer's Responses to Questions

**Comments to the Author**

1. Is the manuscript technically sound, and do the data support the conclusions?

Reviewer #1: Partly

Reviewer #2: Yes

2. Has the statistical analysis been performed appropriately and rigorously? 

Reviewer #1: Yes

Reviewer #2: Yes

3. Have the authors made all data underlying the findings in their manuscript fully available?

Reviewer #1: Yes

Reviewer #2: Yes

4. Is the manuscript presented in an intelligible fashion and written in standard English?

Reviewer #1: Yes

Reviewer #2: Yes

5. Review Comments to the Author

Reviewer #1: This is an interesting paper predicting drug usage in the Guangdong region by analysis of wastewater metabolites. The paper is well written and appears to use methodologies used for other such studies of other regions. The paper should be of interest to researchers employing simar and other techniques to track drug abuse in municipal areas.

(1) One thing that concerns me is the large standard errors reported for the data in Table 5. In most cases the errors are nearly the same magnitude as the numbers themselves. Do these large errors allow clear conclusions to be drawn about trends in drug usage? How do these standard errors compare to other studies of this type? Are they comparable in magnitude, and to be expected?

(2) It would be helpful to have standard errors for the data in Table 6 as well. Why do the numbers in the table have different numbers of significant figures ranging from 1 to 3? Is this warranted and intended?

(3) It appears the data in Figures 1 and 2 is the same as the data presented in tabular form in Tables 4 and 5, is that correct? If so it does not seem including the figures contributes anything additional.

Reviewer #2: Thank you for the opportunity to review your paper titled "Estimation of the prevalence of drug abuse by wastewater-based epidemiology study in four cities of Guangdong, China." Overall I found your study interesting and of benefit to the scientific community. I have several recommendations for improvement:

1. I would recommend replacing the term "drug abuse" with "substance use." While some of the substances that you are measuring may be from abusing drugs you have no way of knowing the circumstances of their consumption. Substance use is a more neutral term.

2. lines 44 and 45. You mention a link between drug abuse and criminal activity. Please provide an explanation and reference.

3. lines 57 - 60. Can you elaborate on how the wastewater analysis works in more detail? From reading further in the paper it seems you are analyzing for metabolites, it would be nice to see that explained in the introduction.

4. line 80 and 319-320. Please elaborate on how your findings can be used to prevent and/or treat substance use. Are you recommending a monitoring program? If so, how would a monitoring program be used to create change?

5. line 288. There is a mention of data from 2021, is this from a different study? Is this a typo?

6. line 197. Why is the prevalence measured for only age 15-64? Can you state why you chose this age group?

7. I noticed a few grammatical/typographical errors. line 74: The word "Therefore" is not needed. line 90: The phrase "In contrast" is not needed. line 93: the word "acquired" is in a different font.

6. PLOS authors have the option to publish the peer review history of their article (what does this mean? ). If published, this will include your full peer review and any attached files.

**Do you want your identity to be public for this peer review?** For information about this choice, including consent withdrawal, please see our Privacy Policy .

Reviewer #1: No

Reviewer #2: No

---

## [Author Response · Author response to Decision Letter 1]

3 Feb 2025

Dear Editors and Reviewers:

Thank you for your letter and for the reviewers’ comments concerning our manuscript entitled “Estimation of the prevalence of drug abuse by wastewater-based epidemiology study in four cities of Guangdong, China” (manuscript ID PONE-D-24-60707). Those comments are all valuable and very helpful for revising and improving our paper, as well as the important guiding significance to our research. We have studied comments carefully and have made corrections which we hope meet with approval. The reviewer comments are laid out below in italicized font and specific concerns have been numbered. Our response is given in normal font and changes/additions to the manuscript are given in blue text. The main corrections in the paper and the responses to the reviewer’s comments are as follows:

1. Response to comment: Please ensure that your manuscript meets PLOS ONE's style requirements, including those for file naming. The PLOS ONE style templates can be found at https://journals.plos.org/plosone/s/file?id=wjVg/PLOSOne_formatting_sample_main_body.pdf and https://journals.plos.org/plosone/s/file?id=ba62/PLOSOne_formatting_sample_title_authors_affiliations.pdf

Response: As suggested by the editor, we have corrected my manuscript to meet PLOS ONE's style requirements.

2. Response to comment: In your Methods section, please provide additional information regarding the permits you obtained for the work. Please ensure you have included the full name of the authority that approved the field site access and, if no permits were required, a brief statement explaining why.

Response: As suggested by the editor, we have added “This study has received approval from the Ethics Committee of the School of Narcotics Control and Public Security, Criminal Investigation Police University of China (2022-12).” in the Methods section.

2. Response to comment: Thank you for stating the following financial disclosure:

“This research was funded by theNational Key R&D Program Project of China (No. 2018YFC0807405)， the Guangdong Provincial Public Security Department Drug Abuse Scale Investigation Project (No. 2019-54).”

Response: As suggested by the editor, we have stated what role the funders took in the study.” The funders had no role in study design, data collection and analysis, decision to publish, or preparation of the manuscript.”

4. Response to comment: PLOS requires an ORCID iD for the corresponding author in Editorial Manager on papers submitted after December 6th, 2016. Please ensure that you have an ORCID iD and that it is validated in Editorial Manager. To do this, go to ‘Update my Information’ (in the upper left-hand corner of the main menu), and click on the Fetch/Validate link next to the ORCID field. This will take you to the ORCID site and allow you to create a new ID or authenticate a pre-existing iD in Editorial Manager.

Response: We're sorry, because the first author's unit, Criminal Investigation Police University of China, does not agree to allow employees to register ORCID ID. Therefore, after a joint discussion among all authors, it has been decided to change the corresponding author and have the author Liu Li from the collaborating unit serve as the corresponding author. This author plays an important role in the project design demonstration, investigation, data analysis, and manuscript writing process. Her ORCID ID will be provided and verified in the editing management system.

5. Response to comment: Please include a separate caption for each figure in your manuscript.

Response: As suggested by the editor, we have added a separate caption for each figure in lines 234-235 and 252-253.” Fig1. Drug residue concentrations (ng/L) of WWTPs in four cities of Guangdong, 2023 to 2024” and “Fig 2. Consumption (mg/1000 inh/d) of drugs of abuse in four cities of Guangdong province, 2023 to 2024”.

6. Response to comment: Please include captions for your Supporting Information files at the end of your manuscript, and update any in-text citations to match accordingly. Please see our Supporting Information guidelines for more information: http://journals.plos.org/plosone/s/supporting-information.

Response: As suggested by the editor, we have included captions for your Supporting Information files at the end of your manuscript, and updated any in-text citations to match accordingly.

7. Response to comment: Please review your reference list to ensure that it is complete and correct. If you have cited papers that have been retracted, please include the rationale for doing so in the manuscript text, or remove these references and replace them with relevant current references. Any changes to the reference list should be mentioned in the rebuttal letter that accompanies your revised manuscript. If you need to cite a retracted article, indicate the article’s retracted status in the References list and also include a citation and full reference for the retraction notice.

Response: As suggested by the editor, we removed the reference 19. Additionally， we replace the reference 45 to 45，46.

19. Thai P.K., O’Brien J.W., Tscharke B.J., Mueller J.F. Analyzing Wastewater Samples Collected during Census To Determine the Correction Factors of Drugs for Wastewater-Based Epidemiology: The Case of Codeine and Methadone. Environ. Sci. Techol. Lett. 2019;6:265–269.

45. Cao Y, Dong X, Shao X, et al. The sewage analysis method monitors the long-term trend of drug abuse in cities. Environmental Science, 2021, 42 (12): 5912-5920.

45.Shao XT, Liu YS, Tan DQ, Wang Z, Zheng XY, Wang DG. Methamphetamine use in typical Chinese cities evaluated by wastewater-based epidemiology. Environ Sci Pollut Res Int. 2020;27(8):8157-8165. doi:10.1007/s11356-019-07504-w

46.Zunong J, Shu MS, Li ML, Asihaer Y, Guan MY, Hu YF. Zhonghua Yu Fang Yi Xue Za Zhi. 2023;57(5):674-678. doi:10.3760/cma.j.cn112150-20221130-01161

Additional Editor Comments:

1. Response to comment: Every figure and table should be able to stand on its own, please make sure that all acronyms and abbreviations are defined in the table or figure legend.

Response: As suggested by the editor, we have made sure that all acronyms and abbreviations are defined in the table or figure legend.

2. Response to comment: I would recommend not using acronyms in the first sentence of the conclusions - this sentence is most commonly quoted, reduce confusion if you use full names.

Response: As suggested by the editor, we have corrected “WBE” to “wastewater-based epidemiology (WBE)”.

3. Response to comment: The first sentence of the conclusion is a bit weak, your real findings are presented on second sentence. consider strengthening the first sentence to include your actual findings.

Response: As suggested by the editor, we have corrected the first sentence to “Utilizing a wastewater-based epidemiology (WBE) approach, this study scientifically evaluates the substance use population's consumption scale and incidence trend in four selected cities of Guangdong Province, China.”

Reviewer #1:

1. Response to comment: One thing that concerns me is the large standard errors reported for the data in Table 5. In most cases the errors are nearly the same magnitude as the numbers themselves. Do these large errors allow clear conclusions to be drawn about trends in drug usage? How do these standard errors compare to other studies of this type? Are they comparable in magnitude, and to be expected?

Response： As suggested by the reviewer, the standard errors reported for the data in Table 5 were large. These standard errors are similar to the research results of Du P, Yargeau V, Postigo C, and others. They are comparable in magnitude, and to be expected reflect on the trend of drug usage. Wastewater-based epidemiology can achieve objective, continuous, and real-time monitoring, which can accurately reflect the drug consumption situation within the community. The advantages include low cost, strong operability, and relatively less time consumption. However, inevitably, there are also shortcomings in wastewater-based epidemiology: there is uncertainty throughout the entire process of wastewater-based epidemiology, including uncertainty in the number of people served by sewage plants, human excretion rates of target substances, substance stability, and concentration of target substances in sewage samples.

Du P, Liu X, Zhong G, et al. Monitoring Consumption of Common Illicit Drugs in Kuala Lumpur, Malaysia, by Wastewater-Cased Epidemiology. Int J Environ Res Public Health. 2020;17(3):889. Published 2020 Jan 31. doi:10.3390/ijerph17030889

Yargeau V, Taylor B, Li H, Rodayan A, Metcalfe CD. Analysis of drugs of abuse in wastewater from two Canadian cities. Sci Total Environ. 2014;487:722-730. doi:10.1016/j.scitotenv.2013.11.094

Postigo C, de Alda ML, Barceló D. Evaluation of drugs of abuse use and trends in a prison through wastewater analysis. Environ Int. 2011;37(1):49-55. doi:10.1016/j.envint.2010.06.012

2. Response to comment: It would be helpful to have standard errors for the data in Table 6 as well. Why do the numbers in the table have different numbers of significant figures ranging from 1 to 3? Is this warranted and intended?

Response： Firstly, as suggested by the reviewer, we have added the standard errors for the data in Table 6. Secondly, we have changed all significant figures in Tables 1-3 to one decimal place.

3. Response to comment: It appears the data in Figures 1 and 2 is the same as the data presented in tabular form in Tables 4 and 5, is that correct? If so it does not seem including the figures contributes anything additional.

Response： As suggested by the reviewer, the data in Figures 1 and 2 is the same as the data presented in tabular form in Tables 4 and 5, but we believe that Figures 1 and 2 can provide a more intuitive view of the differences in drug residue concentration and consumption between the four cities, so it is necessary to retain them.

Reviewer #2:

1. Response to comment: I would recommend replacing the term "drug abuse" with "substance use." While some of the substances that you are measuring may be from abusing drugs you have no way of knowing the circumstances of their consumption. Substance use is a more neutral term.

Response： As suggested by the reviewer, we have replaced the term "drug abuse" with "substance use".

2. Response to comment: lines 44 and 45. You mention a link between drug abuse and criminal activity. Please provide an explanation and reference.

Response： As suggested by the reviewer, we have added the explanation “substance use is associated with antisocial and aggressive behavior” and reference [5].

3. Response to comment: lines 57 - 60. Can you elaborate on how the wastewater analysis works in more detail? From reading further in the paper it seems you are analyzing for metabolites, it would be nice to see that explained in the introduction.

Response： As suggested by the reviewer, we have added the explanation of the wastewater-based epidemiology in lines 55-57. “and found that by using domestic sewage to detect the concentration of cocaine and its metabolites, the estimated per capita consumption of cocaine was highly consistent with the data from previous surveys.”

4. Response to comment: line 80 and 319-320. Please elaborate on how your findings can be used to prevent and/or treat substance use. Are you recommending a monitoring program? If so, how would a monitoring program be used to create change?

Response： As suggested by the reviewer, we have added the explanation of the substance use monitoring program in lines 84-86 and 324-326. “to establish a substance use monitoring program based on sewage epidemiology to detect the temporal trends of major used substances, and to prevent substance use. “ and ”establishing a substance use monitoring program based on wastewater-based epidemiology to detect the temporal trends of major used substances, and to prevent substance use.“

5. Response to comment: line 288. There is a mention of data from 2021, is this from a different study? Is this a typo?

Response： Sorry， this is our mistake， this comes from the same study. We have deleted” Our estimations suggest that in 2021”

6. Response to comment: line 197. Why is the prevalence measured for only age 15-64? Can you state why you chose this age group?

Response： Because people aged 15–64 are considered to be the most important group of drug abusers, the prevalence of drug abusers in people aged 15–64 was calculated by the average amount of abuse, typical dose, and typical frequency. This equation is consistent with the research formula of Haijian Lu et al.

Lu H, Liu W, Zhang H, Yang J, Liu Y, Chen M, Guo C, Sun X, Xu J. Investigation on consumption of psychoactive substances and their ecological risks using wastewater-based epidemiology: a case study on Qinghai-Tibet Plateau. Environ Sci Pollut Res Int. 2023 Feb;30(8):21815-21824. doi: 10.1007/s11356-022-23744-9. Epub 2022 Oct 24. PMID: 36279058.

7. Response to comment: I noticed a few grammatical/typographical errors. line 74: The word "Therefore" is not needed. line 90: The phrase "In contrast" is not needed. line 93: the word "acquired" is in a different font.

Response： As suggested by the reviewer, we have made changes to the corresponding positions in lines 76, 93, and 96 of the manuscript.

We tried our best to improve the manuscript and made some changes in the manuscript. These changes will not influence the content and framework of the paper.

Here we did not list the changes but marked them with different colors in the revised paper.

We appreciate for Editors/Reviewers’ warm work earnestly, and hope that the correction will meet with approval. Once again, thank you very much for your comments and suggestions.

Wish you all the best!

Sincerely yours,

Wei Wang

---

## [Editor Report · Decision Letter 1]

14 Feb 2025

Estimation of the prevalence of substance use by wastewater-based epidemiology study in four cities of Guanmgdong, China

PONE-D-24-60707R1

Dear Dr. Liu,

We’re pleased to inform you that your manuscript has been judged scientifically suitable for publication and will be formally accepted for publication once it meets all outstanding technical requirements.

Kind regards,

David Wampler

Academic Editor

PLOS ONE

Additional Editor Comments (optional):

Thank you, Authors addressed all of the concerns addressed by the reviewers.

Well Done.
---

## [Editor Report · Acceptance letter]

PONE-D-24-60707R1

PLOS ONE

Dear Dr. Liu,

I'm pleased to inform you that your manuscript has been deemed suitable for publication in PLOS ONE. Congratulations! Your manuscript is now being handed over to our production team.

Kind regards,

on behalf of

Dr. David Wampler

Academic Editor

PLOS ONE